# Target-specific membrane potential dynamics of neocortical projection neurons during goal-directed behavior

**Takayuki Yamashita[1,2]\*, Carl CH Petersen[1]\***

[1]Laboratory of Sensory Processing, École Polytechnique Fédérale de Lausanne (EPFL), Lausanne, Switzerland; [2]Department of Neuroscience II, Research Institute of Environmental Medicine, Nagoya University, Nagoya, Japan

**Abstract** Goal-directed behavior involves distributed neuronal circuits in the mammalian brain, including diverse regions of neocortex. However, the cellular basis of long-range cortico-cortical signaling during goal-directed behavior is poorly understood. Here, we recorded membrane potential of excitatory layer 2/3 pyramidal neurons in primary somatosensory barrel cortex (S1) projecting to either primary motor cortex (M1) or secondary somatosensory cortex (S2) during a whisker detection task, in which thirsty mice learn to lick for water reward in response to a whisker deflection. Whisker stimulation in 'Good performer' mice, but not 'Naive' mice, evoked long-lasting biphasic depolarization correlated with task performance in S2-projecting (S2-p) neurons, but not M1-projecting (M1-p) neurons. Furthermore, S2-p neurons, but not M1-p neurons, became excited during spontaneous unrewarded licking in 'Good performer' mice, but not in 'Naive' mice. Thus, a learning-induced, projection-specific signal from S1 to S2 may contribute to goal-directed sensorimotor transformation of whisker sensation into licking motor output.

**\*For correspondence:** takayuki.
yamashita@riem.nagoya-u.ac.jp
(TY); carl.petersen@epfl.ch (CCHP)

**Competing interests:** The authors declare that no competing interests exist.

## Introduction

Primary sensory cortex processes incoming sensory information flexibly in an experience, context and task-dependent manner (*Gilbert and Li, 2013*; *Harris and Mrsic-Flogel, 2013*). Functionally-tuned sensory information is signaled from primary sensory cortex to distinct cortical areas (*Movshon and Newsome, 1996*; *Sato and Svoboda, 2000*; *Chen et al., 2013*; *Glickfeld et al., 2013*; *Yamashita et al., 2013*), but the cellular mechanisms underlying specific cortico-cortical signals during goal-directed behavior are poorly understood.

Neuronal activity in primary somatosensory barrel cortex (S1) is known to participate in the execution of a simple whisker-dependent detection task, in which thirsty mice learn to lick a spout in order to obtain a water reward (*Sachidhanandam et al., 2013*). In well-trained mice, putative excitatory neurons in layer 2/3 of S1, on average, have a long-lasting biphasic depolarization after whisker deflection in hit trials, whereas in miss trials the late depolarization is smaller in amplitude (*Sachidhanandam et al., 2013*). However, there is considerable variability across different recordings (*Sachidhanandam et al., 2013*), which could in part relate to distinct types of excitatory projection neurons. Layer 2/3 of S1 barrel cortex has major anatomical ipsilateral cortico-cortical connections to primary whisker motor cortex (M1) and secondary somatosensory cortex (S2) (*Aronoff et al., 2010*). M1-projecting (M1-p) and S2-projecting (S2-p) neurons in layer 2/3 of S1 are likely to be distinct cell-types exhibiting differential patterns of gene expression (*Sorensen et al., 2015*), distinct intrinsic electrophysiological properties in vivo (*Yamashita et al., 2013*), and carrying functionally different signals (*Sato and Svoboda, 2010*; *Chen et al., 2013*; *2015*; *Yamashita et al., 2013*). Retrograde labeling suggests that M1-p and S2-p neurons in S1 are largely non-overlapping types of excitatory

**eLife digest** Many animals can learn quickly to associate specific behaviors with rewards, such as food. Often, the animal's senses of smell, taste, and touch trigger the behavior, and this allows an animal to respond favorably to changes in its environments. However, it is not clear exactly what happens in the animal's brain to reinforce a behavior that ends in a reward, or how its senses help trigger the rewarding behavior.

Yamashita and Petersen have now studied what happens in the brains of mice that were taught to complete a task to get a reward. In the training, thirsty mice learned that they would receive a reward of water if they licked a water spout after they were briefly touched on one of their whiskers. Then, Yamashita and Petersen measured electrical changes in the brain cells of the trained mice and compared those with measurements from the brain cells of untrained mice. The measurements specifically focused on the brain cells that receive sensory information from the whiskers. These cells are in a region of the brain called the primary sensory cortex, which is known to help mice carry out the task. This brain area in turn sends signals to many downstream areas of the brain.

Yamashita and Petersen found that learning the task appeared to enhance the signaling of some cells in this area of the mouse brain. However, this was only the case for the cells that send signals to a region of the brain that further processes the sensory information (the so-called secondary sensory cortex). Other cells that are intermingled in this region but send signals to the part of the brain that controls movement (the motor cortex) were not affected in this way. Together the data suggest that routing signals from the primary sensory cortex to specific downstream areas might allow animals to learn tasks that depend on responding to sensory cues. More studies are now needed to understand exactly how these signals are generated and whether they contribute to triggering the licking behavior in the mice.

neurons (*Sato and Svoboda, 2010*; *Chen et al., 2013*; *Yamashita et al., 2013*). Here, we investigate the cellular basis of selective signaling of sensorimotor information in distinct cortico-cortical pathways during the whisker detection task through membrane potential ($V_m$) recordings of M1-p and S2-p neurons, finding that task learning induces a licking-related depolarization specifically in S2-p neurons.

## Results

### Differential $V_m$ responses in S2-p and M1-p neurons during task performance

Thirsty mice were trained to lick for water reward in response to a 1 ms deflection of the right C2 whisker (*Sachidhanandam et al., 2013*; *Sippy et al., 2015*), and whole-cell $V_m$ recordings were targeted through two-photon microscopy to fluorescently-labelled M1-p and S2-p neurons in layer 2/3 of the C2 barrel column in S1 of the left hemisphere (*Yamashita et al., 2013*) (*Figure 1A,B*). We used two types of mice for recordings: (1) 'Good performer' mice that exhibited a high discriminability between test trials and catch trials (for details see Materials and Methods) during recordings (59 recordings in 27 mice; hit rate, 0.77 ± 0.03; false alarm rate, 0.17 ± 0.01; d' = 2.12 ± 0.09; d' > 1.1 for each recording; *Figure 1C*) learned through training sessions (typically 7–13 daily sessions prior to the recording day, but some mice learned more quickly), responding with a reaction time of 317 ± 17 ms (time from whisker deflection to tongue contact with the water spout) (*Figure 1D*); and (2) 'Naive' mice that were used for recordings on the first day of being exposed to the task and showed no apparent discrimination (36 recordings in 16 mice; hit rate, 0.31 ± 0.03; false alarm rate, 0.28 ± 0.03; d' = 0.03 ± 0.09; d' < 0.9, for each recording; *Figure 1C*), with a mean reaction time of 369 ± 23 ms which was significantly slower than 'Good performer' mice (p=0.0014; *Figure 1E*).

In trained 'Good performer' mice, whisker stimulation evoked a biphasic $V_m$ depolarization in hit trials for both M1-p and S2-p neurons (*Figure 1D*, *Figure 1—figure supplement 1,3*). The early sensory response was not different comparing M1-p and S2-p neurons (p=0.40), but S2-p neurons had significantly larger $V_m$ depolarization ($\triangle V_m$) during the late phase ($\triangle V_m$ at 0.05 – 0.25 s after whisker

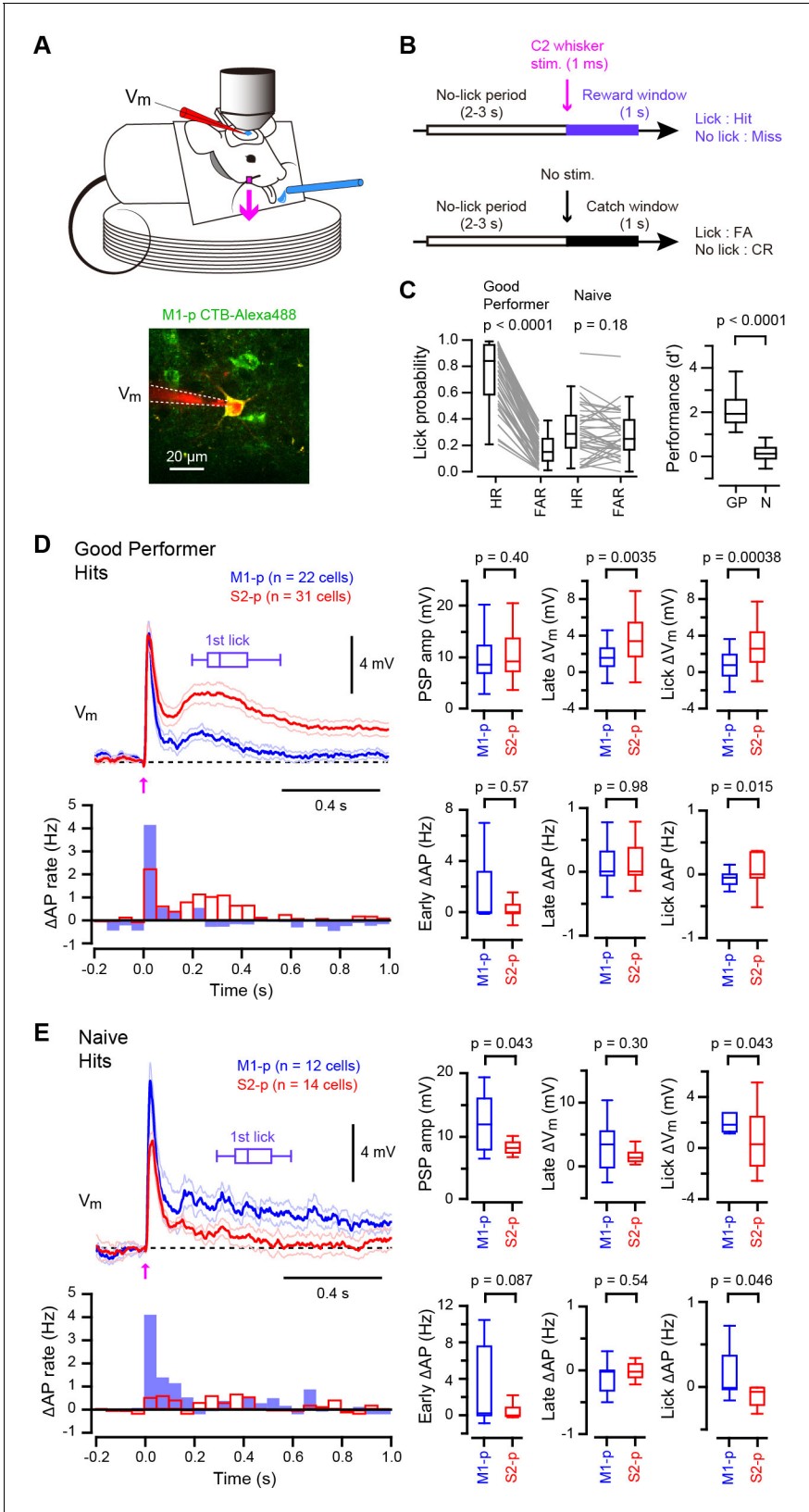

**Figure 1.** Target-specific $V_m$ dynamics in S1 projection neurons during task performance. (**A**) Top, the experimental setup. Bottom, a representative two-photon image of a CTB-labeled M1-p neuron (green) with a recording pipette (red). (**B**) Detection task trial structure (FA: false alarm, CR: correct rejection). (**C**) Behavioral performance during whole-cell recordings (HR: hit rates, FAR: false alarm rates, GP: 'Good performer', N: 'Naive'). (**D**, **E**) Left,

*Figure 1 continued on next page*

*Figure 1 continued*

grand average changes in $V_m$ (thick line: mean, thin lines: ± sem) and action potential (AP) firing rate for hit trials recorded from S2-p neurons (red) and M1-p neurons (blue) in 'Good performer' (D) and 'Naive' (E) mice (Arrow: 1 ms stimulation of the C2 whisker). A box plot indicates reaction time (1st lick). Right, box plots for postsynaptic potential (PSP) amplitude, secondary late $V_m$ depolarization quantified at 0.05–0.25 s, $V_m$ depolarization during the lick period (at 0.25–1.0 s), and evoked AP rates at early (0–0.05 s), late (0.05–0.25 s) and lick (0.25–1.0 s) periods.

The following source data and figure supplements are available for figure 1:

**Source data 1.** Data values and statistics underlying *Figure 1*.

**Source data 2.** Data values and statistics underlying *Figure 1—figure supplement 3*.

**Source data 3.** Data values and statistics underlying *Figure 1—figure supplement 4*.

**Figure supplement 1.** Hit $V_m$ traces from S1 projection neurons in 'Good performer' mice.

**Figure supplement 2.** Hit $V_m$ traces from S1 projection neurons in 'Naive' mice.

**Figure supplement 3.** Average hit $V_m$ traces and PSTHs.

**Figure supplement 4.** Target-specific changes of hit responses with task learning.

deflection: S2-p=4.00 ± 0.59 mV, n = 31; M1-p=1.68 ± 0.44 mV, n = 22; p=0.0035) and during the subsequent lick period ($\triangle V_m$ at 0.25 – 1.0 s after whisker deflection: S2-p=3.10 ± 0.50 mV, n = 31; M1-p=0.73 ± 0.30 mV, n = 22; p=0.00038) (*Figure 1D*, *Figure 1—figure supplement 1,3*). The evoked action potential (AP) rate of S2-p neurons compared to M1-p neurons was also significantly higher during the lick period (p=0.015), but not during early (p=0.57) or late (p=0.98) response periods (*Figure 1D*, *Figure 1—figure supplement 1,3*).

In randomly licking 'Naive' mice (*Figure 1C*), M1-p neurons, compared to S2-p neurons, exhibited larger postsynaptic potentials (PSPs) in response to whisker stimulation in hit trials (PSP amplitude: S2-p=8.41 ± 0.69 mV, n = 14; M1-p=12.20 ± 1.25 mV, n = 12; p=0.043) and larger depolarizations during the licking phase ($\triangle V_m$ at 0.25 – 1.0 s after whisker deflection: S2-p=0.45 ± 0.62 mV, n = 14; M1-p=2.19 ± 0.57 mV, n = 12; p=0.043) (*Figure 1E*, *Figure 1—figure supplement 2,3*). The AP rates in 'Naive' mice during the lick period were also significantly larger in M1-p neurons compared to S2-p neurons (p=0.046) (*Figure 1E*, *Figure 1—figure supplement 2,3*).

Therefore, analyzed for hit trials, S2-p neurons were more strongly excited during licking compared to M1-p neurons in 'Good performer' mice, but, interestingly, the opposite was true for 'Naive' mice in which M1-p neurons were more excited during licking compared to S2-p neurons. Notably, the secondary long-lasting depolarization in S2-p neurons after whisker stimulation was seen only in 'Good performer' mice, not in 'Naive' mice (*Figure 1D,E*, *Figure 1—figure supplement 4*), while the small sustained depolarization of M1-p neurons in 'Naive' mice was attenuated in 'Good performer' mice (*Figure 1D,E*, *Figure 1—figure supplement 4*).

## Differential hit *vs* miss responses in S2-p and M1-p neurons

We next examined whether the $V_m$ dynamics of S1 projection neurons correlated with task execution on a trial-by-trial basis. In S2-p neurons of 'Good performer' mice, the amplitude of PSPs and the late $\triangle V_m$ were slightly larger in hit compared to miss trials (PSPs increased by 20%, p=0.026; late $\triangle V_m$ increased by 39%, p=0.029) (*Figure 2A*, *Figure 2—figure supplement 1* and *2*). Furthermore, the $\triangle V_m$ in S2-p neurons during the licking period was substantially larger (270% increase) in hit compared to miss trials ($\triangle V_m$ at 0.25 – 1.0 s: hit 2.63 ± 0.55 mV, miss 0.71 ± 0.47 mV, n = 19, p=0.0014) (*Figure 2A*, *Figure 2—figure supplement 1* and *2*). Thus, the $V_m$ dynamics of S2-p neurons after whisker deflection were correlated with task execution in trained mice. However, hit and miss trials were not significantly different in M1-p neurons of 'Good performer' mice in early (p=0.23), late (p=0.43) or licking (p>0.99) phases (*Figure 2B*, *Figure 2—figure supplement 1* and *2*).

In contrast, for 'Naive' mice, S2-p neurons did not distinguish hit and miss trials in early (p=0.24), late (p=0.67) or licking (p=0.71) response phases (*Figure 2C*, *Figure 2—figure supplement 1* and

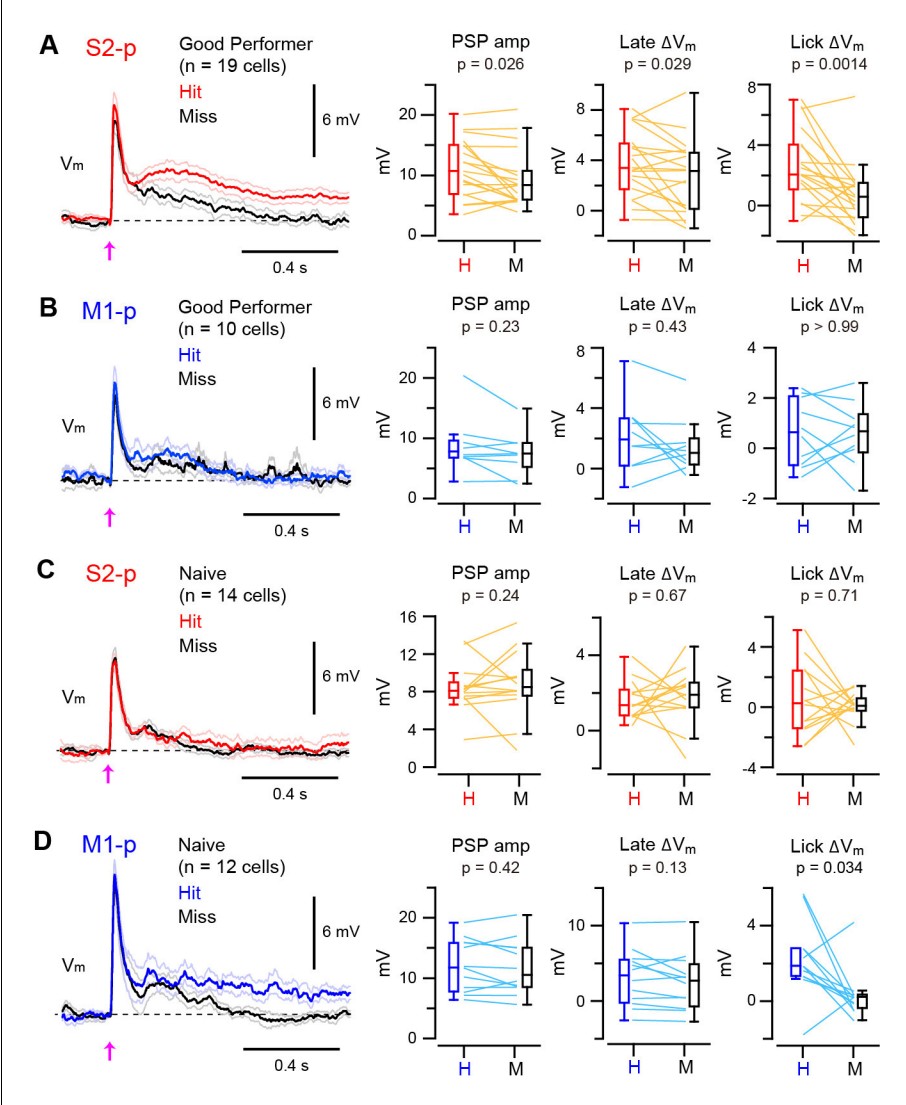

**Figure 2.** Target-specific $V_m$ correlation with task execution. (A) Left, grand average $V_m$ traces (thick line: mean, thin lines: ± sem) of S2-p neurons during hit (red) and miss (black) trials for 'Good performer' mice. Right, data for each cell (thin lines) and box plots for PSP amplitude and $V_m$ depolarization at the late (0.05–0.25 s) and lick periods (0.25–1.0 s) on hit (H) and miss (M) trials. (B) Same as A, but for M1-p neurons. (C) Same as A, but for S2-p neurons in 'Naive' mice. (D) Same as C, but for M1-p neurons.

The following source data and figure supplements are available for figure 2:

**Source data 1.** Data values and statistics underlying *Figure 2*.

**Source data 2.** Data values and statistics underlying *Figure 2—figure supplement 2*.

**Figure supplement 1.** Representative hit/miss $V_m$ traces.

**Figure supplement 2.** Mouse-by-mouse analysis of hit/miss responses.

*2*). M1-p neurons in 'Naive' mice also had similar hit and miss responses during early (p=0.42) and late (p=0.13) periods. However, M1-p neurons in 'Naive' mice had significantly larger $\triangle V_m$ during the lick period in hit trials compared to misses ($\triangle V_m$ at 0.25 – 1.0 s: hit 2.19 ± 0.57 mV, miss 0.30 ± 0.38 mV, n = 12, p=0.034) (*Figure 2D*, *Figure 2—figure supplement 1*).

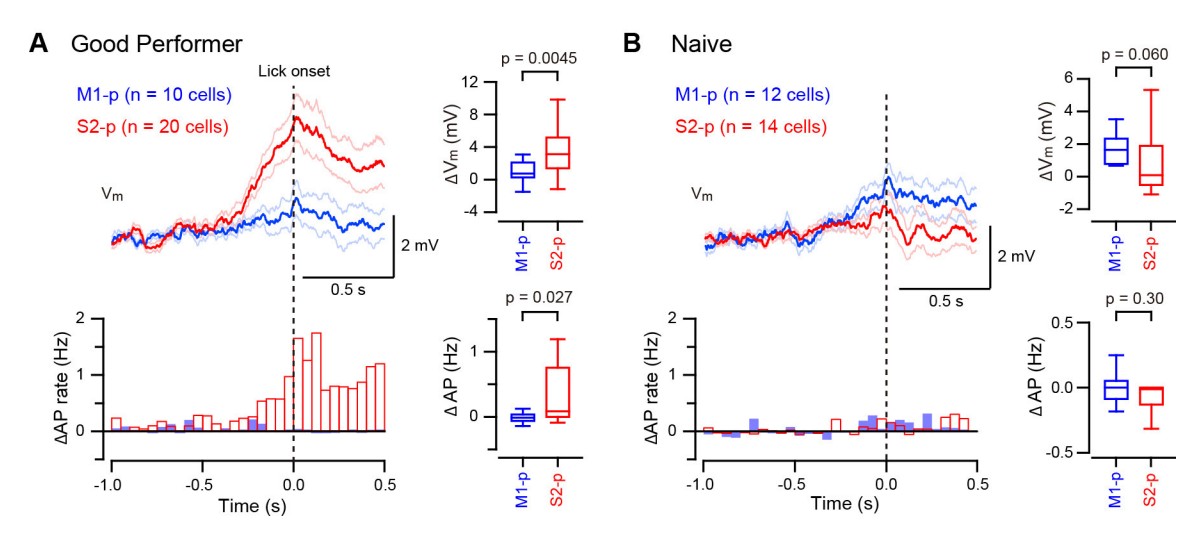

**Figure 3.** Target-specific $V_m$ depolarization during spontaneous unrewarded licking. (**A, B**) Left, grand average change in $V_m$ (thick line: mean, thin lines: ± sem) and AP rate aligned at the onset of detected spontaneous licking (dotted line) in M1-p and S2-p neurons of 'Good performer' (**A**) and 'Naive' (**B**) mice. Right, quantifications at ± 0.1 s around the detected lick onset.

The following source data and figure supplements are available for figure 3:

**Source data 1.** Data values and statistics underlying *Figure 3*.

**Source data 2.** Data values and statistics underlying *Figure 3—figure supplement 3*.

**Figure supplement 1.** Licking-related $V_m$ dynamics of S2-p and M1-p neurons in 'Good performer' mice.

**Figure supplement 2.** Licking-related $V_m$ dynamics of S2-p and M1-p neurons in 'Naive' mice.

**Figure supplement 3.** Further analysis of $V_m$ dynamics during spontaneous unrewarded licking.

Thus, S2-p neurons, but not M1-p neurons, in 'Good Performer' mice had larger depolarizing responses in hit trials compared to misses, whereas in 'Naive' mice M1-p neurons, but not S2-p neurons, had a larger depolarization during licking in hit trials compared to misses.

## Differential depolarization of S2-p and M1-p neurons during spontaneous licking

Some S2-p neurons depolarized strikingly during spontaneous unrewarded licking (*Figure 3—figure supplement 1*). We therefore examined licking-related $V_m$ dynamics and found that S2-p neurons of 'Good performer' mice depolarized during spontaneous unrewarded licking, peaking at around the time when the tongue first contacted the water spout, ($\triangle V_m$ at ± 0.1 s around tongue-spout contact: 3.48 ± 0.62 mV, n = 20). Licking-related depolarization was significantly (p=0.0045) smaller in M1-p neurons of 'Good performer' mice ($\triangle V_m$: 0.83 ± 0.44 mV, n = 10) (*Figure 3A*, *Figure 3—figure supplement 1,3*). S2-p neurons of 'Good performer' mice also increased firing rate significantly during licking compared to M1-p neurons (p=0.027). Licking-related $V_m$ and AP modulation was weak in 'Naive' mice, and it was not significantly different comparing S2-p and M1-p neurons ($\triangle V_m$, p=0.060; $\triangle AP$, p=0.30) (*Figure 3B*, *Figure 3—figure supplement 2,3*).

S2-p neurons, but not M1-p neurons, in 'Good performer' mice are therefore excited during spontaneous licking, whereas in 'Naive' mice there was little spontaneous licking-related activity in S2-p or M1-p neurons. The licking-related depolarization in S2-p neurons was significantly larger in 'Good performer' mice compared to that in 'Naive' mice (*Figure 3—figure supplement 3*), suggesting emergence of projection-specific excitation related to licking after task learning.

## Discussion

Our projection-specific $V_m$ measurements in mice with different levels of task proficiency suggest that cortico-cortical signals originating from S1 are bi-directionally modulated by task learning in a pathway-specific manner (*Figure 4*). In 'Naive' mice, whisker stimulation evoked the strongest signals in M1-p neurons during hit trials, whereas in 'Good performer' mice S2-p neurons showed the strongest excitation during hit trials. The largest differences in activity during task performance between S2-p and M1-p neurons were observed during the lick period, and task learning was accompanied by enhanced excitation during spontaneous licking specifically in S2-p neurons.

The lack of task-correlated activity in M1-p neurons in 'Good performer' mice is consistent with results from a previous study of a closely-related whisker detection task in which inactivation of whisker M1 did not reduce hit rates in trained animals, but rather increased false-alarm rates (*Zagha et al., 2015*). Thus signals from S1 to whisker M1 may not be essential for task execution.

Optogenetic inactivation of S1 during the late phase impairs task performance (*Sachidhanandam et al., 2013*), suggesting a causal role for late excitation. S2-p neurons exhibited a learning-induced depolarization at the late and lick phases of hit trials and during spontaneous unrewarded licking. The grand-averaged, late depolarization in S2-p neurons on hit trials peaked at 261 ms after whisker stimulation (*Figure 1D*), which was earlier than the mean reaction time (317 ms). The licking-related depolarization in S2-p neurons started shortly (260 ± 46 ms, n = 18) before tongue-spout contact during spontaneous licking (*Figure 3A*), which is consistent with the larger depolarization at early and late phases of their responses in hit compared to miss trials (*Figure 2A*). Interestingly, S2 has been suggested to be reciprocally connected to a tongue/jaw-related M1/M2 area (also termed anterior lateral motor cortex, ALM) a neocortical region known to be involved in goal-directed licking (*Oh et al., 2014*; *Guo et al., 2014*; *Li et al., 2015*). We therefore speculate that the licking-related signals in S2-p neurons in S1 might contribute to exciting neurons in tongue/jaw-related M1/M2 via S2 through reciprocally connected networks of excitatory long-range projection neurons, thus contributing to driving licking motor output (*Figure 4*). Consistent with such a hypothesis involving reciprocal excitation between S1 and S2, axons from S2 innervating S1 were found to exhibit strong task-related hit *vs* miss modulation in a closely-related whisker detection task (*Yang et al., 2016*).

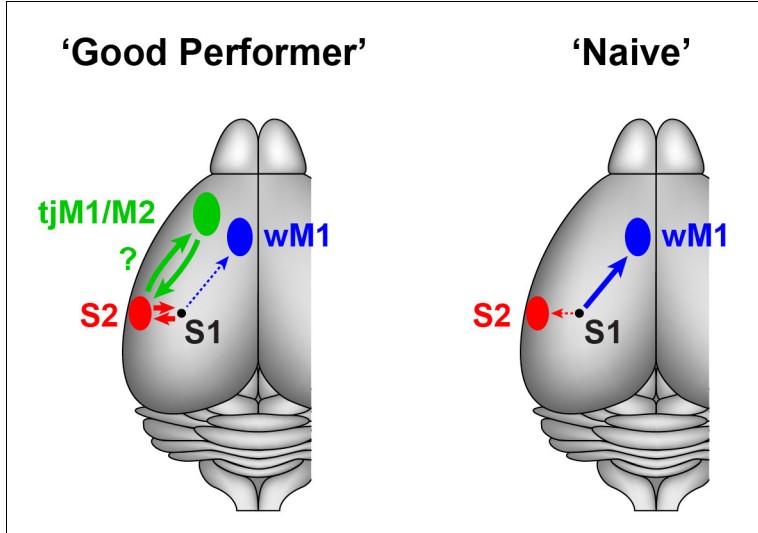

**Figure 4.** Schematic summary of target-specific output signals from S1 during execution of the whisker detection task, together with speculative hypothesis relating to possible cortico-cortical signalling pathways from S1 to tongue/jaw area of motor cortex. Left, In 'Good performer' mice, S2-p neurons, not M1-p neurons, in S1 develop depolarization correlated with task performance. The activities of S2-p neurons could be routed toward the tongue/jaw area of M1/M2 (tjM1/M2). Right, In 'Naive' mice, M1-p neurons, not S2-p neurons, exhibit depolarization correlated with task execution. wM1: whisker M1.

It is important to note that there are many possible sensory/motor signals that might contribute to the lick-related depolarization of S2-p neurons in trained mice: movement of jaw and tongue must begin before tongue-spout contact, and we did not quantify orofacial movements during task performance. Furthermore, rodents are known to have reward-expecting orofacial movements such as sniffing and whisking (*Deschênes et al., 2012*). However, transection of the facial motor nerve that controls whisker movements has no impact on task performance or the late phase $V_m$ (*Sachidhanandam et al., 2013*), suggesting that the late phase $V_m$ might be generated by internal brain circuits rather than sensory reafference coming from associated whisker movements. In this study we primarily compare neurons projecting to different targets in mice with the same level of task proficiency (i.e. S2-p *vs* M1-p in 'Good performer' mice, or S2-p *vs* M1-p in 'Naive' mice), and the differences found comparing these projection neurons can therefore not reflect differences in sensorimotor behavior. In future experiments, it will be important to examine causal roles of S2-p neurons, as well as to investigate the synaptic mechanisms driving the target-specific $V_m$ dynamics in M1-p and S2-p neurons associated with task learning.

## Materials and methods

All experimental procedures were approved by the Swiss Federal Veterinary Office.

### Animal preparation

Implantation of a metal head-restraint post on male C57BL6J mice (6-week-old or older), identification of the locations of the S1-C2 barrel column and whisker-S2 of the left hemisphere by intrinsic optical signal imaging, and the injection of CTB conjugated with Alexa-Fluor 488 or 594 (0.5% in PBS, weight/volume, Invitrogen) into left whisker M1 (1 mm anterior, 1 mm lateral from Bregma) and left S2 were performed as previously described (*Yamashita et al., 2013*). The injection volume of CTB was 50 – 100 nl for M1 and 25 – 50 nl for S2 at the depths of 300 and 800 μm, giving a total volume of 100 – 200 nl for M1 and 50 –100 nl for S2. Animals were kept with a light/dark cycle (12 hr/12 hr) in cages of four mice or less. Experiments were typically performed during the dark period.

### Detection task training

At least one day after CTB injection, mice started to be water-restricted. The mice were adapted to head restraint on the recording setup through initial training to freely lick the water spout for receiving water reward (3 – 5 sessions, one session per day). Mice were then taught to associate whisker deflection with water availability through daily training sessions, essentially as described previously (*Sachidhanandam et al., 2013*; *Sippy et al., 2015*). For whisker stimulus, we used a brief (1 ms) magnetic pulse to elicit a vertical deflection of the right C2 whisker transmitted by a small metal particle glued on the whisker. The reward time window was 1 s after the whisker stimulus throughout training. Trials with whisker stimulation (test trials) or those without whisker stimulation (catch trials) were started without preceding cues at random inter-trial intervals ranging from 2 – 10 s. Catch trials were randomly interleaved with test trials, with 40 – 50% probability of all trials. If the mouse licked in the 2 s (or 3 s in some experiments) preceding the time when the trial was supposed to occur, then the trial was aborted. Catch trials were present from the first day of training. After each training session, 1.0 – 1.5 g of wet food pellet was given to the mouse in order to keep its body weight more than 80% of the initial value. Behavioral control and behavioral data collection were carried out with custom-written computer routines using a National Instruments board interfaced through LabView.

### Whole-cell recordings in task-performing mice

Whole-cell patch-clamp recordings (95 recordings in total) were targeted to cell bodies of CTB-labeled neurons in the center of the C2 barrel column (as identified with intrinsic optical signal imaging) of adult C57BL6J mice (8-week-old or older) under visual control using a custom-built two-photon microscope, as previously described (*Yamashita et al., 2013*). Recordings were made at the subpial depth of 120 – 270 μm, and the recording depths for M1-p and S2-p neurons were similar. The recording pipettes had resistances of 5 – 7 MΩ and were filled with a solution containing (in mM): 135 potassium gluconate, 4 KCl, 10 HEPES, 10 sodium phosphocreatine, 4 MgATP, 0.3

Na$_3$GTP (adjusted to pH 7.3 with KOH). For targeting CTB-labeled neurons, Alexa 488 or 594 (1 – 20 μM) was added to the pipette solution, depending on the color of the targeted cells. In most experiments, we targeted either M1-p or S2-p neurons. In one mouse, we injected CTB-Alexa 488 in M1 and CTB-Alexa 594 in S2 and targeted both M1-p and S2-p neurons. The V$_m$ was measured using a MultiClamp 700B amplifier with a 10 kHz low pass Bessel filter, and digitized at 20 kHz by a National Instruments board. V$_m$ was not corrected for liquid junction potential.

Short (1 min) sweeps of the V$_m$ and the behavioral signals from the lick sensor together with TTL signals to control the water valve and the electromagnetic coil were recorded using Ephus in Matlab (*Suter et al., 2010*). We used two types of mice for recordings: (1) 'Good performer' mice that exhibited a high discriminability between test trials and catch trials during recordings (59 recordings in 27 mice; hit rate, $0.77 \pm 0.03$; false alarm rate, $0.17 \pm 0.01$; d' = $2.12 \pm 0.09$) learned through training sessions (typically 7–13 daily sessions prior to the recording day, but some mice learned more quickly); and (2) 'Naive' mice that were used for recordings on the first day of being exposed to the task and showed no apparent discrimination (36 recordings in 16 mice; hit rate, $0.31 \pm 0.03$; false alarm rate, $0.28 \pm 0.03$; d' = $0.03 \pm 0.09$; d' < 0.9, for each recording). For calculating d' when hit rate or false alarm rate was measured as 1.0 or 0.0, each value was corrected by subtracting or adding $1/(2N)$, where $N$ is the trial number. Each recording typically lasted ~20 min or less, and we made multiple whole-cell recordings from one animal in most of the experiments. For each recording we routinely monitored the level of task performance by calculating d' and discarded data with d' < 1.1 in 'Good performer' mice or those with d' > 0.9 in 'Naive' mice. The d' values for recordings of M1-p and S2-p neurons was not significantly different (p=0.73; for 'Good performer' mice; p=0.50 for 'Naive' mice).

## Data analysis

Subthreshold postsynaptic potentials (PSPs) were analyzed after removing APs by median-filtering (*Crochet and Petersen, 2006*). For analysis of V$_m$ changes evoked by task-relevant whisker deflection, baseline V$_m$ was defined as the mean V$_m$ at 0 – 5 ms before the stimulus onset. The amplitude of PSPs was defined as the difference between the baseline V$_m$ and the peak V$_m$ of averaged traces. The $\triangle$V$_m$ at the late and lick periods was estimated as the difference between the baseline V$_m$ and the mean V$_m$ of the averaged traces at 0.05 – 0.25 s (late) or 0.25 – 1.0 s (lick) after whisker stimulus. APs evoked by whisker stimulation were estimated by subtracting spontaneous AP rate from the AP rate measured in the early (0 – 0.05 s), late (0.05 – 0.25 s) or lick (0.25 –1.0 s) periods after the stimulation for each cell. Baseline AP rates were computed as the mean of no-lick periods (2 s before test/catch trials) totaling over 16 s. Peristimulus time histograms (PSTHs) were computed by counting AP number in each 50 ms (or 10 ms) bin for each cell and averaging the number across cells recorded. Grand average PSTHs are shown in Hz after subtracting baseline AP rates. On average, 31 $\pm$ 2 hit trials (n = 53 cells) and 18 $\pm$ 2 miss trials (n = 29 cells) per recording were analyzed for 'Good performer' mice, and 14 $\pm$ 1 hit trials (n = 26 cells) and 31 $\pm$ 3 miss trials (n = 26 cells) per recording were analyzed for 'Naive' mice. In some recordings the well-trained mouse showed few misses and in such cases we only analyzed hit responses.

Lick bouts that occurred at least 3 s after whisker stimulation, and at least 1 s after the cessation of previous lick bouts were selected for analysis of V$_m$ modulation induced by spontaneous unrewarded licking. On average, 64 $\pm$ 6 lick bouts (n = 30 cells) of 'Good performer' mice and 31 $\pm$ 3 lick bouts (n = 26 cells) of 'Naive' mice were analyzed for each recording. The individual V$_m$ traces aligned at the onset of detected lick bouts (lick onset) were median-filtered to remove APs. Baseline V$_m$ was defined as the mean V$_m$ at 1.0 – 0.6 s before the lick onset, and the magnitude of V$_m$ modulation was estimated by the difference between the baseline V$_m$ and the mean V$_m$ at $\pm$ 0.1 s around the lick onset. APs evoked during lick events were calculated by subtracting baseline AP rate (averaged at 0.6 – 1.0 s before the detected lick onset) from the AP rate measured within $\pm$ 0.1 s from the lick onset. PSTHs around lick events are shown in Hz after subtracting the baseline AP rate. The onset of the licking-related V$_m$ depolarization was computed as the time point where V$_m$ increased over 3 x SD of the baseline V$_m$ for the 18 out of 20 S2-p cells with pre-lick depolarization.

All values (except for box plots) are presented as mean $\pm$ sem. Box plots indicate median and 1st/3rd quartile, with Tukey's whiskers showing maximal and minimal data points within 1.5 times interquartile range away from 1st/3rd quartile. Statistical testing using two-tailed Wilcoxon rank-sum test for unpaired data (for example, M1-p *vs* S2-p for *Figure 1D,E* and *Figure 3A,B*; 'Good

performer' *vs* 'Naive' for *Figure 1C*) and two-tailed Wilcoxon signed rank test for paired data (for example, hit *vs* miss trials for *Figure 2*; hit *vs* false-alarm rates for *Figure 1C*) was performed in Igor-Pro (WaveMetrics) without excluding any data points. Testing for the normality of data distribution was performed in IgorPro and we found that at least one of the samples in every two-sample comparison was not normally distributed. Non-parametric tests were therefore used for all figures. We analyzed data on a cell-by-cell basis unless otherwise noted. Neither randomization nor blinding was done for data collection or analysis.

## Acknowledgements

We thank Taro Kiritani and Varun Sreenivasan for help with building the experimental setup, and Eloïse Charrière for help with mouse training. This work was supported by grants from the Swiss National Science Foundation and the European Research Council.

## Additional information

### Funding

| Funder | Grant reference number | Author |
| --- | --- | --- |
| Schweizerischer Nationalfonds zur Förderung der Wissenschaftlichen Forschung | 310030_146252 | Carl CH Petersen |
| European Research Council | 293660 | Carl CH Petersen |

The funders had no role in study design, data collection and interpretation, or the decision to submit the work for publication.

### Author contributions

TY, Designed the project, Wrote the manuscript, Performed experiments, Analyzed data, Conceptionand design, Acquisition of data, Analysis and interpretation of data, Drafting or revising the article; CCHP, Designed the project, Wrote the manuscript, Conception and design, Analysis and interpretation of data, Drafting or revising the article

### Author ORCIDs

Takayuki Yamashita, http://orcid.org/0000-0002-1849-9837
Carl CH  Petersen, http://orcid.org/0000-0003-3344-4495

### Ethics

Animal experimentation: All experimental procedures were approved by the Swiss Federal Veterinary Office, authorisation VD1628.

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
