## [Decision Letter]

Thank you for submitting your article "Target-specific membrane potential dynamics of neocortical projection neurons during goal-directed behavior" for consideration by *eLife*. Your article has been favorably evaluated by Timothy Behrens (Senior editor) and three reviewers, one of whom, Naoshige Uchida, is a member of our Board of Reviewing Editors.

The reviewers have discussed the reviews with one another and the Reviewing Editor has drafted this decision to help you prepare a revised submission.

Summary:

The authors performed whole-cell recording from neurons in the primary somatosensory cortex (S1) while mice performed a whisker deflection detection task. The main finding is that S2-projecting, but not M1-projecting, neurons show long-lasting depolarization at the late phase (0.05 – 0.25s) after whisker stimulation. Furthermore, this difference was observed only in well-trained mice but not in naïve mice. Overall, all the reviewers thought that the data are highly valuable, and that the findings are of great interest. However, it was pointed out that there are some interpretational difficulties. Also, some statistical analyses require more explicit justifications and explanations.

Essential revisions:

1) It was discussed that "the licking-related signals in S2-p neurons in S1 might contribute to exciting neurons in tongue/jaw-related M1 via S2 through reciprocally connected networks of excitatory long-range projection neurons, thus contributing to driving licking motor output (Figure 4)". While this is an interesting possibility, alternative possibilities are not considered. What if the learning-related changes in membrane potential dynamics are secondary to changes in behavior? For instance, in addition to expected changes in licking behavior, other orofacial movements and whisking might change during learning. More importantly, sensation associated with these behaviors may change during learning and cause alterations in membrane potential dynamics. If the altered membrane potential dynamics are caused by these secondary changes, the interrelation will be very different. Ideally, these issues should be addressed by additional experiments or measurements. For instance, the authors might record from S2-projecting neurons while paralyzing the whisker pad and removing whisker movement (or whisker pad movement). We understand that this experiment may take more than two months, and you might decide not to do these experiments. If you choose not to perform these experiments, these alternative interpretations must be discussed in the revised manuscript very carefully and in a very visible and balanced way.

2) Animal-to-animal differences in behavior/learning may cause biases in neuronal responses. The authors need to make a convincing argument that neurons from the same mice can be treated independently. More details need to be provided on the statistical analysis and how these confounding factors may have been dealt with.

Reviewer #1:

In the manuscript entitled "Target-specific membrane potential dynamics of neocortical projection neurons during goal-directed behavior", Yamashita and Petersen performed whole-cell recording from a large number of neurons (n = 95 neurons from at least 27 mice) in the primary somatosensory cortex (S1) while mice performed a whisker deflection detection task. Notably, whole cell recording was targeted to neurons retrogradely labeled from their projection target (secondary somatosensory cortex [S2] and primary motor cortex). Furthermore, the authors also compared the data from well-trained (after 7-13 days of training) versus naïve (the first day of training) mice.

The authors found that S2-projecting, but not M1-projecting, neurons show long-lasting depolarization at the late phase (0.05 – 0.25s) after whisker stimulation. Furthermore, this difference was observed only in well-trained mice but not in naïve mice. The authors also found that S2-projecting neurons in well-trained mice show depolarization immediately before spontaneous, unrewarded licking. In naïve mice, M1-projecting neurons in naïve mice showed stronger response than S2-projecting neurons. The authors discuss that S2-projecting S1 neurons may be responsible in generating licking response in well-trained mice, and the activation of S2-projecting neurons before licking may be due to the reciprocal connection between S2 and the tongue/jaw-related area in M1.

The authors collected a large data set and the conclusions are convincing. The manuscript is written clearly and the conclusions are very interesting. I have a few relatively minor comments.

1) The definition of "good performer" and "naïve" mice is done only in the Materials and methods section. It would be very helpful if the authors explain this in the main text when they appear.

2) The histogram in Figure 1 indicates that S2-projecting neurons show on average a higher rate of firing during the late response phase than M1-projecting neurons. However, the box plot does not show any difference. First, is there any difference in the data between these plots? Second, this result may suggest that action potentials sent to the postsynaptic neurons are not different between the two populations. Is it possible that whole cell recording artificially reduced the firing rate responses in recorded neurons?

3) Are there S1 neurons that project to both S2 and M1? How much would this affect the results?

Reviewer #2:

This paper is part of a substantial body of work from the Petersen lab, investigating L2/3 networks in the barrel cortex in the context of tactile passive detection tasks. Yamashita and Petersen (2013) previously reported that S1 neurons projecting to primarymotor cortex (M1) and those projecting to secondary somatosensory cortex (S2) have distinct intrinsic membrane properties and exhibit different membrane potential dynamics during spontaneous behavior. Here, by making whole-cell recordings in primary somatosensory, they compare these two projection classes in the context of a simple tactile detection behavior. They also compare behavior-related activity in the projection classes in naive and performing mice.

They make interesting observations. In expert mice, S2-p neurons show larger slow depolarizations after whisker deflection, and also depolarize with licking. In contrast, in naive mice, M1-p neurons show larger deflections. So there is a change in the response properties with learning, suggesting changing routing of signals with learning. This is hard-won data. Whole cell recordings are rarely applied to behaving mice, much less in a projection-specific manner. The experimental approach is pioneering. However, there are some weaknesses that make the data difficult to interpret. It's not clear to me is there is a change in routing with learning; the mouse might instead employ distinct sensorimotor strategies, which recruit the different projection classes in a different manner. It's nice to see differences in coding across projection classes, but the implications beyond this interesting observation are unclear.

1) What might the learning-related changes mean? The paper is silent on any interpretation. Is the idea that some stimulus-response association is coded in the network (whisker – S2p – tongue M1)? Does Figure 4 suggest that functional connectivity changes to recruit different loops? If so how? How about the alternative explanation that S2-p neurons report orofacial sensation related to movement (which differs for different training conditions). As is, it seems that the behavioral design does not allow separating these and other interpretations.

2) In the 2013 paper differences between M1p and S2p neurons are dramatic. Here the differences are subtle. In particular, the data in Figure 2 (trial-trial analysis) appears marginal. This is somewhat worrisome given some uncertainties with stats.

3) Related: It's not clear to me if the stats were done properly. They need to convince me that neurons from the same mice can be treated independently. This is critical here because behavior was not monitored (for example, whisker movement) and there likely are strong correlations across neurons of one animal, imposed by idiosyncratic behavior. In addition, it's not clear that S2p and M1p projecting neurons were recorded in the same proportions in each animal etc. A lot more details needs to be provided on the statistical analysis and how these confounding factors may have been dealt with.

4) Finally, it's a bit surprising that whisker movements, details of tongue movements, and learning-related changes thereof, where not monitored. How do we know that behavior (learning; hit vs. miss) are not associated with different movements and sensory input related to movement? Clearly the licking will be very different in expert mice compared to naive mice. Similar for other orofacial movements.

5) What is the implication of the data of Figure 3. Is this a sensory or motor signal? Does activity precede the earliest signs of movement?

Other points:

Why does the 'lick period' extend long after the all-important first lick?

Reviewer #3:

This is a brief paper from Yamashita and Petersen on the change in intracellular response on neurons in vS1 that project to vS2 versus vM1. The authors find differences – a greater change in vS2 versus vM1 projecting cells – which is of importance in deciphering the rules of neuronal plasticity in cortex and in explaining the variance observed in prior experiments with functionally unlabeled cells. This submission extends earlier work from the Petersen laboratory (Sachidhanandam et al. Nature Neuroscience 2013 and Yamashita et al. Neuron 2013) as well as the Helmchen laboratory (Chen et al. Nature 2013, Chen et al. Nature Neuroscience 2015). The conclusions of the present work are similar to those in Chen 2015. With no slight toward the Chen 2015 paper, I find the present work more compelling as it is based on intracellular measurements, which reveals significant structure about the nature of the response, as opposed to calcium-dynamics in Chen 2015, which is largely a means to report bursts of spikes.

The authors present a wealth of intracellular data on the response to deflection of a vibrissa in trained versus naive animals. There are a number of important differences in the records in the data that are unreported, so I would ask the authors to revise the Results section to address the following.

1) With respect to Figure 1 – the effect of "good performer" versus "naive" appears to be mainly a substantial increase in the depolarization of S2-projecting cells as opposed to a change in M1-projecting cells, whose response is largely unchanged (compare parts D and E). The text describes only the difference in the relative depolarization; I think this statement can be strengthened.

2) With respect to the depolarization for the "good performers", there is a significant delayed peak at about 250 ms into the trial. This is never discussed, but should be; it seems like a long time for a recurrent signal.

In a real sense, this paper and the Chen 2015 complement each other. The large calcium imaging set in Chen 2015 allow those authors to delineate an increase in discrimination by S2 projecting neurons (Chen 2015 Figure 5D and 6), while the present work shows the nature of the change in subthreshold response.

---

## [Author Response]

Essential revisions:

1) It was discussed that "the licking-related signals in S2-p neurons in S1 might contribute to exciting neurons in tongue/jaw-related M1 via S2 through reciprocally connected networks of excitatory long-range projection neurons, thus contributing to driving licking motor output (Figure 4)". While this is an interesting possibility, alternative possibilities are not considered. What if the learning-related changes in membrane potential dynamics are secondary to changes in behavior? For instance, in addition to expected changes in licking behavior, other orofacial movements and whisking might change during learning. More importantly, sensation associated with these behaviors may change during learning and cause alterations in membrane potential dynamics. If the altered membrane potential dynamics are caused by these secondary changes, the interrelation will be very different. Ideally, these issues should be addressed by additional experiments or measurements. For instance, the authors might record from S2-projecting neurons while paralyzing the whisker pad and removing whisker movement (or whisker pad movement). We understand that this experiment may take more than two months, and you might decide not to do these experiments. If you choose not to perform these experiments, these alternative interpretations must be discussed in the revised manuscript very carefully and in a very visible and balanced way.

This is an important point. The “licking-related” signal in S2-p neurons of Good Performer mice could have various origins including both sensory and motor signals. In a previous study of the same detection task, we carried out facial motor nerve section, and this had no impact on licking behavior or the late phase V_m_ (Sachidhanandam et al., 2013). Furthermore, in the same study, we also ascribed a causal role of the late activity in S1 for our detection task, through temporally-precise optogenetic inactivation during the late phase (Sachidhanandam et al., 2013). We therefore think that it is not inappropriate to speculate that the licking-related signal in S2-p neurons might contribute to task performance by enhancing the lick probability.

However, we cannot rule out the possibility that orofacial sensorimotor activity contributes to the licking-related signal in S2-p neurons, and indeed it is possible that some orofacial sensory-reafference signals are important for task performance, and for generating the licking-related signal in S2-p neurons. In the Discussion section we now write:

“Optogenetic inactivation of S1 during the late phase impairs task performance (Sachidhanandam et al., 2013), suggesting a causal role for late excitation. […] In future experiments, it will be important to examine causal roles of S2-p neurons, as well as to investigate the synaptic mechanisms driving the target-specific V_m_ dynamics in M1-p and S2-p neurons associated with task learning.”

2) Animal-to-animal differences in behavior/learning may cause biases in neuronal responses. The authors need to make a convincing argument that neurons from the same mice can be treated independently. More details need to be provided on the statistical analysis and how these confounding factors may have been dealt with.

This is an interesting point, and the reviewer is correct to point out that animal-to-animal differences could contribute to some aspects of our dataset.

We therefore pooled data of multiple cells for each mouse and analysed the V_m_ and APs during task performance for S2-p and M1-p neurons. Our main conclusions that S2-projecting, but not M1-projecting, neurons in trained mice show long-lasting depolarization at the late phase after whisker stimulation (Figure 1—figure supplement 3), that hit trials are more depolarized than miss trials during late and licking periods for S2-p neurons (Figure 2—figure supplement 2), and licking-related depolarization (Figure 3—figure supplement 3) remain prominent and statistically significant even when analysed on a mouse-by-mouse basis. Our results are therefore robust.

For main figures, we would like to keep analysis on a cell-by-cell basis. The reasons for this are as follows: (1) In our data set, there is no clear evidence that there is stronger correlation across neurons of one animal over correlation across animals; (2) The number of cells we recorded in one animal is small (2 cells on average), so animal-to-animal differences, if any, could not effectively affect results; (3) The cell-to-cell variability of membrane potential properties such as spike rates is high in these projection neurons (Yamashita et al., Neuron, 2013), presumably overwhelming animal-to-animal variability in our experimental condition; (4) As far as we know all published papers using similar techniques (for example, Cohen et al., Nature 2012; Haider et al., Nature 2013; Polack et al., Nature Neuroscience, 2013; Zagha et al., Neuron 2015; Yang et al., Nature Neuroscience, 2016) also treat cells independently.

We also added more detailed information on how we collected and analyzed data in the Materials and methods section of the revised manuscript.

Reviewer #1:

[…] The authors collected a large data set and the conclusions are convincing. The manuscript is written clearly and the conclusions are very interesting. I have a few relatively minor comments.

1) The definition of "good performer" and "naïve" mice is done only in the Materials and methods section. It would be very helpful if the authors explain this in the main text when they appear.

We have now added statements at the beginning of the Results section to explain the definition of "good performer" and "naive" mice. We write: “We used two types of mice for recordings: (1) ‘Good performer’ mice that exhibited a high discriminability between test trials and catch trials (for details see Materials and methods) during recordings (59 recordings in 27 mice; hit rate, 0.77 ± 0.03; false alarm rate, 0.17 ± 0.01; d’ = 2.12 ± 0.09; d’ > 1.1 for each recording; Figure 1) learned through training sessions (typically 7–13 daily sessions prior to the recording day, but some mice learned more quickly), responding with a reaction time of 317 ± 17 ms (time from whisker deflection to tongue contact with the water spout) (Figure 1); and (2) ‘Naive’ mice that were used for recordings on the first day of being exposed to the task and showed no apparent discrimination (36 recordings in 16 mice; hit rate, 0.31 ± 0.03; false alarm rate, 0.28 ± 0.03; d’ = 0.03 ± 0.09; d’ < 0.9, for each recording; Figure 1), with a mean reaction time of 369 ± 23 ms which was significantly slower than ‘Good performer’ mice (p = 0.0014; Figure 1).”

2) The histogram in Figure 1 indicates that S2-projecting neurons show on average a higher rate of firing during the late response phase than M1-projecting neurons. However, the box plot does not show any difference. First, is there any difference in the data between these plots? Second, this result may suggest that action potentials sent to the postsynaptic neurons are not different between the two populations. Is it possible that whole cell recording artificially reduced the firing rate responses in recorded neurons?

The grand average V_m_ and PSTH (Figure 1 left) and the box plots (Figure 1 right) show different analyses of the same data. For the box plots we divide the post-stimulus timeline into three epochs: early (0 – 50 ms), late (50 – 250 ms) and licking (250 – 1000 ms). We found larger V_m_ depolarization during the late and licking periods, and enhanced AP firing in the licking period for S2-p neurons compared to M1-p neurons. In our view, the box plots are consistent with the PSTH.

So far across our different studies, whole-cell recordings have generated similar data to cell-attached recordings (Gentet et al., 2010; Sachidhanandam et al., 2013).

3) Are there S1 neurons that project to both S2 and M1? How much would this affect the results?

There are very few layer 2/3 S1 neurons double-labeled with retrograde tracers injected into S2 and M1 (0.7% , Yamashita et al., 2013). We are in the process of reconstructing the full axonal arborisations of M1-p and S2-p neurons, and, so far across our small sample of 15 neurons, we have not found any that project to both targets. On P. 3 we now write: “Retrograde labeling suggests that M1-p and S2-p neurons in S1 are largely non-overlapping types of excitatory neurons (Sato and Svoboda, 2010; Chen et al., 2013; Yamashita et al., 2013).”

*Reviewer #2:*

*This paper is part of a substantial body of work from the Petersen lab, investigating L2/3 networks in the barrel cortex in the context of tactile passive detection tasks. Yamashita and Petersen (2013) previously reported that S1 neurons projecting to primarymotor cortex (M1) and those projecting to secondary somatosensory cortex (S2) have distinct intrinsic membrane properties and exhibit different membrane potential dynamics during spontaneous behavior. Here, by making whole-cell recordings in primary somatosensory, they compare these two projection classes in the context of a simple tactile detection behavior. They also compare behavior-related activity in the projection classes in naive and performing mice.*

*They make interesting observations. In expert mice, S2-p neurons show larger slow depolarizations after whisker deflection, and also depolarize with licking. In contrast, in naive mice, M1-p neurons show larger deflections. So there is a change in the response properties with learning, suggesting changing routing of signals with learning. This is hard-won data. Whole cell recordings are rarely applied to behaving mice, much less in a projection-specific manner. The experimental approach is pioneering. However, there are some weaknesses that make the data difficult to interpret. It's not clear to me is there is a change in routing with learning; the mouse might instead employ distinct sensorimotor strategies, which recruit the different projection classes in a different manner. It's nice to see differences in coding across projection classes, but the implications beyond this interesting observation are unclear.*

1) What might the learning-related changes mean? The paper is silent on any interpretation. Is the idea that some stimulus-response association is coded in the network (whisker – S2p – tongue M1)? Does Figure 4 suggest that functional connectivity changes to recruit different loops? If so how? How about the alternative explanation that S2-p neurons report orofacial sensation related to movement (which differs for different training conditions). As is, it seems that the behavioral design does not allow separating these and other interpretations.

We agree with the reviewer that the functional meaning of the learning-induced changes in V_m_ is unclear. Our results are consistent with the possibility that the licking-related depolarization in S2-p neurons might be routed to tongue/jaw M1 via S2. However, the reviewer is correct to point out that alternative explanations could be possible. We made this point clearer by adding a question mark in Figure 4, and adding discussion: “It is important to note that there are many possible sensory/motor signals that might contribute to the lick-related depolarization of S2-p in trained mice: movement of jaw and tongue must begin before tongue-spout contact, and we did not quantify orofacial movements during task performance. […] In this study we primarily compare neurons projecting to different targets in mice with the same level of task proficiency (i.e. S2-p vs M1-p in ‘Good performer’ mice, or S2-p vs M1-p in ‘Naive’ mice), and the differences found comparing these projection neurons can therefore not reflect differences in sensorimotor behavior.”

2) In the 2013 paper differences between M1p and S2p neurons are dramatic. Here the differences are subtle. In particular, the data in Figure 2 (trial-trial analysis) appears marginal. This is somewhat worrisome given some uncertainties with stats.

In this study we found many robust and prominent differences between M1-p and S2-p neurons. The differences in late and lick V_m_ (Figure 1) and spontaneous lick-related V_m_ changes (Figure 3) are obvious and have very high statistical significance. The hit/miss differences in Figure 2 are also robust, and are also found when analysed on a mouse-by-mouse manner (see below). Indeed, each of the three main figures shows data with statistical significance P < 0.01.

3) Related: It's not clear to me if the stats were done properly. They need to convince me that neurons from the same mice can be treated independently. This is critical here because behavior was not monitored (for example, whisker movement) and there likely are strong correlations across neurons of one animal, imposed by idiosyncratic behavior. In addition, it's not clear that S2p and M1p projecting neurons were recorded in the same proportions in each animal etc. A lot more details needs to be provided on the statistical analysis and how these confounding factors may have been dealt with.

This is an interesting point, and the reviewer is correct to point out that animal-to-animal differences could contribute to some aspects of our dataset.

We therefore pooled data of multiple cells for each mouse and analysed the V_m_ and APs during task performance for S2-p and M1-p neurons. Our main conclusions that S2-projecting, but not M1-projecting, neurons in trained mice show long-lasting depolarization at the late phase after whisker stimulation (Figure 1—figure supplement 3), that hit trials are more depolarized than miss trials during late and licking periods for S2-p neurons (Figure 2—figure supplement 2), and licking-related depolarization (Figure 3—figure supplement 3) remain prominent and statistically significant even when analysed on a mouse-by-mouse basis. Our results are therefore robust.

For main figures, we would like to keep analysis on a cell-by-cell basis. The reasons for this are as follows: (1) In our data set, there is no clear evidence that there is stronger correlation across neurons of one animal over correlation across animals; (2) The number of cells we recorded in one animal is small (2 cells on average), so animal-to-animal differences, if any, could not effectively affect results; (3) The cell-to-cell variability of membrane potential properties such as spike rates is high in these projection neurons (Yamashita et al., Neuron, 2013), presumably overwhelming animal-to-animal variability in our experimental condition; (4) As far as we know all published papers using similar techniques (for example, Cohen et al., Nature 2012; Haider et al., Nature 2013; Polack et al., Nature Neuroscience, 2013; Zagha et al., Neuron 2015; Yang et al., Nature Neuroscience, 2016) also treat cells independently.

We also added more detailed information on how we collected and analyzed data in the Materials and methods section of the revised manuscript.

4) Finally, it's a bit surprising that whisker movements, details of tongue movements, and learning-related changes thereof, where not monitored. How do we know that behavior (learning; hit vs. miss) are not associated with different movements and sensory input related to movement? Clearly the licking will be very different in expert mice compared to naive mice. Similar for other orofacial movements.

The “licking-related” signal could have many origins including both sensory and motor signals. We have already carried out facial motor nerve section, and this had no impact on behavior or the late phase V_m_ (Sachidhanandam et al., 2013). The projection-specific nature of the signal is seen for the same type of mice (Good performer or Naive), and a large part of our story is thus independent of what correlated changes in behavior take place over learning. We added discussion on this point in the Discussion section (last paragraph), see our reply to point 1.

5) What is the implication of the data of Figure 3. Is this a sensory or motor signal? Does activity precede the earliest signs of movement?

The licking-related depolarization in S2p neurons starts slightly before tongue- spout contact. In a previous study we found that optogenetic inactivation of S1 during the late period caused deficits in the same detection task (Sachidhanandam et al., 2013). The licking-related activity of S2-p neurons could therefore contribute as a motor-related signal. However, there could be movements of tongue and other orofacial parts that occur earlier than tongue- spout contact and give rise to sensory-reafference signals. It is therefore not possible to make a definitive statement based on our current data. We added discussion on this point in the revised manuscript (last paragraph), see our reply to point 1.

*Other points:*

Why does the 'lick period' extend long after the all-important first lick?

The “lick period” covers the licking period to measure licking-related signals. The “late” period on the other hand precedes the first lick.

Reviewer #3:

This is a brief paper from Yamashita and Petersen on the change in intracellular response on neurons in vS1 that project to vS2 versus vM1. The authors find differences – a greater change in vS2 versus vM1 projecting cells – which is of importance in deciphering the rules of neuronal plasticity in cortex and in explaining the variance observed in prior experiments with functionally unlabeled cells. This submission extends earlier work from the Petersen laboratory (Sachidhanandam et al. Nature Neuroscience 2013 and Yamashita et al. Neuron 2013) as well as the Helmchen laboratory (Chen et al. Nature 2013, Chen et al. Nature Neuroscience 2015). The conclusions of the present work are similar to those in Chen 2015. With no slight toward the Chen 2015 paper, I find the present work more compelling as it is based on intracellular measurements, which reveals significant structure about the nature of the response, as opposed to calcium-dynamics in Chen 2015, which is largely a means to report bursts of spikes.

The authors present a wealth of intracellular data on the response to deflection of a vibrissa in trained versus naive animals. There are a number of important differences in the records in the data that are unreported, so I would ask the authors to revise the Results section to address the following.

1) With respect to Figure 1 – the effect of "good performer" versus "naive" appears to be mainly a substantial increase in the depolarization of S2-projecting cells as opposed to a change in M1-projecting cells, whose response is largely unchanged (compare parts D and E). The text describes only the difference in the relative depolarization; I think this statement can be strengthened.

To show the learning-induced changes in membrane potential dynamics of projection neurons, we added new supplementary figures (Figure 1—figure supplement 4 and Figure 3—figure supplement 3) comparing between naive and good-performer mice (i.e., M1p naive vs. M1p experts and S2p naive vs. S2p expert).

We accordingly added statements on this, as suggested (Results):

“Notably, the secondary long-lasting depolarization in S2-p neurons after whisker stimulation was seen only in ‘Good performer’ mice, not in ‘Naive’ mice (Figure 1, Figure 1—figure supplement 4), while the small sustained depolarization of M1-p neurons in ‘Naive’ mice was attenuated in ‘Good performer’ mice (Figure 1, Figure 1—figure supplement 4).”

“The licking-related depolarization in S2-p neurons was significantly larger in ‘Good performer’ mice compared to that in ‘Naive’ mice (Figure 3—figure supplement 3), suggesting emergence of projection-specific excitation related to licking after task learning.”

2) With respect to the depolarization for the "good performers", there is a significant delayed peak at about 250 ms into the trial. This is never discussed, but should be; it seems like a long time for a recurrent signal.

The long-lasting depolarization for “good performers” is quantified across late (50 – 250 ms period after whisker stimulation) and licking (250 – 1000 ms) periods. The peak of the grand average V_m_ trace is at 261 ms, which precedes licking (mean reaction time was 317 ms). As suggested, we now added discussion on the peak time of the late depolarization:

“S2-p neurons exhibited a learning-induced depolarization at the late and lick phases of hit trials and during spontaneous unrewarded licking. […] We therefore speculate that the licking-related signals in S2-p neurons in S1 might contribute to exciting neurons in tongue/jaw-related M1/M2 via S2 through reciprocally connected networks of excitatory long-range projection neurons, thus contributing to driving licking motor output (Figure 4).”

The timescale is indeed long, but even simple reciprocal long-range feedback interactions can take place on long time-scales. For example, in the study of Manita et al. (2015) they find ~100 ms in an S1->M2->S1 loop. We think it is likely that there might be more complex interactions across different areas in our behavior, perhaps involving both S1, S2 and tjM1, which could give rise to even longer timescales. The mechanisms driving slow, behaviorally relevant membrane potential depolarization are indeed of enormous interest to investigate in the future.

In a real sense, this paper and the Chen 2015 complement each other. The large calcium imaging set in Chen 2015 allow those authors to delineate an increase in discrimination by S2 projecting neurons (Chen 2015 Figure 5D and 6), while the present work shows the nature of the change in subthreshold response.

We agree with the reviewer.